# Exploring the Feasibility of Intrapartum GBS Collection to Identify Residual GBS in a Pilot Study of an Antenatal Probiotic Intervention

Emily Malloy [1,2,*], Lisa Hanson [1,*], Leona VandeVusse [1], Karen Robinson [1], Maharaj Singh [1] and Marie Forgie [2]

1   College of Nursing, Marquette University, Milwaukee, WI 53233, USA
2   Department of Obstetrics and Gynecology & Midwifery and Wellness, Advocate Aurora UW Medical Group, Aurora Sinai Medical Center, Milwaukee, WI 53233, USA
*   Correspondence: emily.malloy@marquette.edu (E.M.); lisa.hanson@marquette.edu (L.H.);
    Tel.: +1-414-288-3841 (L.H.)

**Abstract:** (1) Background: We aimed to explore the feasibility of collecting intrapartum maternal Group B Streptococcus (GBS) colonization and immediate post-birth neonatal GBS colonization cultures for use in a larger trial and to identify cases of residual GBS, which were hypothesized to be less common in the probiotics group. (2) Methods: This sub-study added additional outcome measures to the parent study to identify intrapartum and neonatal colonization and compare between probiotic and placebo groups and to identify cases of residual GBS. Intrapartum maternal vaginal and rectal GBS cultures were collected at the time of admission to a hospital for labor and to give birth. Neonatal oral and nasopharynx GBS cultures were collected within 1–2 h of giving birth. (3) Results: Thirty intrapartum samples were collected; twenty-eight had complete data. The antepartum GBS results significantly predicted the intrapartum results ($p = 0.005$), with 86.7% of cultures remaining the same at both time points. There were four cases where the intrapartum GBS results were different to the 36-week antepartum cultures results. A case of residual GBS was identified in one probiotic group participant. None of the neonatal swabs were positive for GBS. No cases of EOGBSD occurred in infants born to the study participants. (4) Conclusions: Although the 36–37 week GBS results significantly predicted the intrapartum results, the utility for a larger research trial on probiotics to reduce antenatal GBS is unclear. Intrapartum GBS swab collection was feasible in a busy nurse, midwife, and physician practice. GBS was not recovered from neonatal oral and nasopharyngeal swabs. The pathways of neonatal GBS colonization require further study.

**Keywords:** group B streptococcus; GBS; probiotics; early-onset GBS disease

## 1. Introduction

*Group B Streptococcus* (GBS), or Streptococcus agalactiae, is a Gram-positive bacterium that is both a leading cause of neonatal infectious morbidity and mortality and a commensal present in the gastrointestinal (GI) and genitourinary (GU) tracts of up to 65% of people and 10–40% of pregnant people worldwide [1–4]. It is implicated in perinatal health problems, including urinary tract infections, premature labors, stillbirths, intraamniotic infections, and endometritis [1].

In the United States of America (USA), the rate of antenatal GBS colonization is approximately 25% (CDC, 2018). The worldwide average colonization rate is 18%, with regional differences having a range of 10–40% [2]. In 2019, the US guidelines for universal screening from the Center from Disease Control (CDC) were transferred to the American College of Obstetricians and Gynecologist (ACOG) and American Academy of Pediatrics (AAP), with the recommendation to screen all pregnant people between 36-weeks and 0 days gestation and 37 weeks 6 days gestation for GBS via vaginal-to-rectal cultures [1]. This change is

reflective of studies that have shown that GBS colonization is usually persistent, but may be transitory, and screening is more accurate if it is performed within 5 weeks of giving birth [3]. According to these Early-Onset GBS Disease (EOGBSD) secondary prevention guidelines, pregnant people colonized with GBS receive intrapartum antibiotic prophylaxis to prevent EOGBSD in their newborns. EOGBSD occurs within 7 days of birth and most commonly from 12 to 48 h of giving birth [1]. The morbidity for neonates includes sepsis, bacteremia, pneumonia, and meningitis, and the mortality rate is approximately 4–8% [4,5].

The purpose of this pilot sub-study was to explore the feasibility of intrapartum maternal GBS collection and immediate post-birth neonatal GBS collection for use in a larger future clinical trial. We also sought to identify cases of residual GBS. We hypothesized that (1) more participants in the probiotics group would test negative for GBS via the intrapartum vaginal-to-rectal swabs compared to those in the placebo group, and (2) fewer cases of residual GBS would be found in probiotic group participants. We also sought to explore the results of post-birth neonatal oral and nasopharynx cultures for collecting GBS compared to those born to control group participants.

Background. According to the US Centers for Disease Control and Prevention (CDC), a pregnant person who screens positive for GBS and receives intrapartum antibiotic prophylaxis has a 1 in 4000 chance of having a neonate who will develop EOGBSD, which is compared to a 1 in 200 chance for a pregnant person who screens positive for GBS and does not receive intrapartum antibiotic prophylaxis [4]. Neonatal GBS colonization occurs following fetal contact with the birth canal during normal labor and a vaginal birth. It is notable that EOGBSD is highly unlikely following a cesarean birth with the amniotic sac intact [1]. EOGBSD is diagnosed when GBS is recovered from an otherwise sterile substance, such as blood or cerebrospinal fluid [6]. Without intrapartum antibiotic prophylaxis, 50% of neonates born to GBS positive mothers will be colonized by GBS, and 1–2% will develop EOGBSD [2]. Recent epidemiologic data from the US show that the rate of EOGBSD is 0.23–0.37 per 1000 live births, while the rate of the Late Onset GBS disease (LOGBDS) is 0.31 per 1000 live births [7]. The exact mechanism/s of neonatal colonization is poorly understood. Healthy full-term and premature infants are not routinely screened for GBS. One study of 160 mother–neonatal pairs tested multiple maternal and neonatal body sites including the vagina, rectum, and breast milk of mothers and the throat and rectum of neonates, and neonates were considered to be GBS colonized if one positive site was positive [8]. Neonatal and maternal GBS strains were identical in every positive case.

Infants with LOGBSD commonly present with bacteremia, meningitis, and occasionally soft tissue or organ infection between 7 days and 3 months of life [1]. LOGBSD is sometimes attributed to breastmilk or horizontal transmission from the birthing parent, but additional sources may include neonatal intensive care unit exposure or from individuals within the household or community [1,8,9]. There are currently no known strategies to prevent LOGBSD.

In the United States, the guidelines for laboratory testing for GBS were managed by the CDC from 1996 to 2019, when the American Society for Microbiology (ASM) began managing the guidelines for GBS-related laboratory practices [10]. The ASM details the optimal collection of swabs, swab type, transportation, storage, interpretation, and antibiotic sensitivity testing of GBS [10]. There are ten documented GBS serotypes [11]. GBS is usually identified by culturing, in which a specific microbe is grown and identified in laboratory media [1,12]. Any GBS colony-forming units (CFUs) identified during pregnancy are considered a positive GBS result.

Residual GBS. Besides exposing 20–30% of pregnant people to two or more doses of intrapartum antibiotics, the CDC/ACOG EOGBSD prevention guideline has an important limitation. Between 61.4% and 81% of term infants with EOGBSD are born to pregnant people who screened negative for GBS during antenatal screening from 35 to 37 weeks' gestation [10,13,14]. This may be due to the transitory nature of GBS colonization, resulting in the conversion from negative to positive after 36 weeks, light maternal colonization, and/or laboratory errors. The term 'residual GBS' was operationally defined for this study

as cases in which a participant was GBS-culture-negative at 36 weeks gestation, but tested GBS-positive on admission because they were in labor and to give birth. Residual GBS is referred to in the literature using a variety of terms, including false-negative screening and missed cases [13,15–18].

Probiotics Used to Reduce Antenatal GBS Colonization. Midwives have suggested using probiotics as a primary prevention strategy for antennal GBS colonization [15,16]. Probiotics are live microorganisms that, when taken in adequate amounts, can lead to health benefits [17,18]. Probiotic bacteria survive the upper gastrointestinal (GI) tract and colonize the lower GI tract and vagina. Probiotic bacteria colonize the mucosal surfaces of these sites and stimulate the production of acids and other substances that prevent the adherence of pathogens like GBS. Nine clinical trials of probiotics to reduce GBS have been published [16,19–26]. The probiotic interventions tested showed antagonistic activity against GBS through several mechanisms of action, including the increased acidification of mucosal surfaces, immune modulation, and decreasing adhesion of bacteria [27].

Intrapartum GBS cultures were used as the primary study outcome in only one study of probiotics to reduce GBS colonization [25]. In this study, 110 Taiwanese pregnant participants who were GBS positive at 35–37 weeks received either a probiotic (*Lactobacillus rhamnosus* GR-1 $\times$ $10^9$ CFU and *Lactobacillus* RC-14 $\times$ $10^9$ CFU) or a placebo. The probiotic intervention significantly reduced intrapartum GBS colonization (21/49 participants, 42.9%) compared to that of the placebo group (9/50, 18.0%) (Chi-square $p = 0.007$).

Hanson and colleagues conducted a systematic review and meta-analysis of ten studies (five in vitro studies and six clinical trials—one with both an in vitro and in vivo arm) that studied the use of an oral probiotic intervention to reduce antenatal GBS colonization [27]. The probiotic intervention was started in the third trimester in all five clinical trials ($n = 705$). The meta-analysis revealed that the use of an antenatal probiotic decreased the probability of a positive GBS result by 44% (OR = 0.56, 95% CI = [0.34, 0.92], $p = 0.02$) [27]. Subsequently, Menichini and colleagues identified five RCTs ($n = 583$) that met the inclusion criteria for their systematic review and meta-analysis. The researchers found that the GBS colonization rate in the probiotics group was 31.9% (96/301) compared to 38.6% (109/282) in the control group (OR = 0.62, 95% CI = [0.40–0.94], I2 4.8%, $p = 0.38$) [28]. They found that when the probiotics intervention was initiated in the third trimester, a greater reduction of GBS colonization was identified ($p = 0.55$).

The safety of the use of probiotic interventions during low-risk pregnancy has been well established. A meta-analysis evaluating the safety of pre- and probiotics during pregnancy and lactation did not identify any serious health risks [29].

## 2. Materials and Methods

### 2.1. Relationship of the Pilot Sub-Study to Parent Double-Blind, Placebo-Controlled RCT

The pilot sub-study added an exploratory and feasibility intrapartum arm on to an NIH-funded, double-blind, randomized, placebo-controlled trial that was in process, "The Efficacy of Probiotics to Reduce Antepartum GBS" (R21HD095320, clinical.trials.gov Identifier: NCT03696953), which will be referred to as the parent study. The methods and findings of the parent study were published in detail [21]. The aim of the parent study was to demonstrate the efficacy of an oral probiotic intervention to reduce GBS colonization at 36 weeks gestation, and therefore, lessen the need for intrapartum antibiotic prophylaxis. Participants were randomly assigned to either Florajen3 probiotic capsules (*Lactobacillus acidophilus*, *Bifidobacterium lactis*, and *Bifidobacterium longum*) in a microcrystalline cellulose carrier (MCC) or a placebo (MCC) that was identical in taste and appearance. The study capsules were initiated at 28 $\pm$ week gestation and taken once daily at the time of labor and giving birth. The primary study outcome was GBS colonization based on positive Standard-of-Care (SOC) GBS vaginal-to-rectal swabs obtained at 36 weeks gestation. Participants in the parent study remained on the probiotic or placebo until the time of birth. The final measure in the parent study was GBS colonization at 36–37 weeks gestation. To account for

participant dropout, up to 116 participants were enrolled to achieve the sample size of 83 (39 probiotic and 43 placebo).

This pilot sub-study was IRB approved (Identifier: NCT03696953) in the final year of the parent study, with a goal of recruiting the remaining 30 participants and 30 neonates. Due to the variable nature of labor and birth onset, there was variation in the time from antenatal GBS testing to the time of labor and giving birth with a range of 0–5 weeks. The results of the parent study found that the probiotic intervention reduced antenatal GBS colonization by 5% ($p = 0.7$) and more significantly reduced gastrointestinal symptoms after 8 weeks of the intervention compared to that of the controls [21].

Following IRB approval for the additional sub-study measures, all remaining participants in the parent study, double-blind, randomized controlled trial (DB-RCT) of antenatal probiotics to reduce GBS consented to the pilot sub-study with an intrapartum arm. Using G*power version 3.1 (Christian-Albrechts-Universität, Kiel, Germany) an estimated a sample of 25 pregnant participants was necessary to have a power of 0.80 or more with a large effect size ($f^2 = 0.35$) and with an alpha of 0.05 for applying linear multiple regression with 3 predictors [30]. All members of the research team, except the Investigational Pharmacist (who prepared the study bottles), were blind to the group assignment. Double blinding was maintained until all data collection was completed and after the parent study was unblinded to group assignment.

### 2.2. Data Collection

Figure 1 contains a schema of the sub-study. Upon admission for labor and to give birth or for the induction of labor, vaginal-to-rectal GBS swabs were collected per parent study protocol by a midwife, physician, or the participant themselves upon presentation to the labor and birth unit. The sub-study GBS swabs were not collected while the participant was in labor, and the ACOG GBS guidelines were followed for IAP during labor based on the 36-week results. The hospital laboratory was used to study the 36-week standard-of-care GBS cultures and the intrapartum vaginal-to-rectal and neonatal cultures. We used a culture-based method, and any GBS CFUs present was considered a positive result. Neonatal nasal and oropharynx swabs were collected by the nurse, midwife, physician, or birth parent within 1–2 h of birth. All study participant and neonatal swabs for GBS were labelled and sent to the hospital laboratory for analysis, and the results appeared in the participant's electronic medical record accessible by the certified research coordinator (CRC). Demographic and perinatal outcome data collected for the parent study were retrieved from the participant electronic medical records and recorded by the CRC using REDCap (Research Electronic Data Capture) electronic data capture tools hosted by Advocate Aurora Healthcare, and then imported into SAS and SPSS [31,32].

| | Baseline 28±2 weeks | 36±2 weeks | Admitted for labor | 1-2 hours post Birth |
|---|---|---|---|---|
| | **Probiotic Capsules**<br><br>**or Placebo Capsules** | | | |
| | Enrollment<br>Consent<br>Allocation<br>Baseline Measures<br>Study bottle | | | |
| GBS Screening Vaginal to Rectal Swab | | Determined need for IAP | **X** | |
| Neonatal nasopharynx GBS swab | | | | **X** |

**Figure 1.** Sub-study schema. Note: IAP = intrapartum antibiotic prophylaxis. X = sub-study GBS measure.

*2.3. Statistical Analysis*

Following the completion of data analysis for the parent study, analysis began in the sub-study. An intention-to-treat model was used for the analysis in the sub-study. All categorical variables, including demographics and perinatal outcomes, were reported as frequencies and percentages and were compared using Chi-square testing or Fishers exact test for two groups (probiotics and control). For numeric continuous variables, the mean for two groups was found using *t* test for independent samples. Logistic regression was used to predict intrapartum GBS. For all statistical tests, an alpha of 0.05 was used as the level of statistical significance. All statistical analysis was conducted using SAS 9.4 SAS Institute, Cary, NC, USA and IBM SPSS, version 28 (IBM Corporation, Armonk, NY, USA).

**3. Results**

Thirty participants were enrolled in the pilot sub-study, and two participants had missing data. One gave birth precipitously (both maternal and neonatal swab missing), and one had an unplanned cesarean birth (neonatal swab missing). Therefore, complete data were available for 28 sub-study participants. The sub-study participant demographics and perinatal outcomes are presented in Table 1. The intrapartum GBS outcomes were compared between probiotic and placebo group participants and are presented in Table 2. There were no differences between the groups for either demographic data or perinatal outcomes.

The logistic regression showed that the standard-of-care swabs significantly predicted the intrapartum GBS results (OR = 33.33, 95% CI = [2.8, 392.5], *p* = 0.005). SOC vaginal-to-rectal 36-week swabs were mismatched with the intrapartum results in 4/30 cases (13.3%). One probiotic participant converted from GBS positive to negative, and one converted from negative to positive. Two placebo group participants converted from positive to negative. No GBS was recovered on any neonatal oral or nasopharyngeal swabs. This feasibility sub-study showed that midwives are able to collect intrapartum GBS swabs (29/30) and that no participants declined the intrapartum swabs.

**Table 1.** Demographic data of participants and perinatal outcomes.

| Variable | Total | Probiotics (*n* = 14) | Placebo (*n* = 16) | *p* Value |
|---|---|---|---|---|
| **Age** (Mean, SD) | 29.87 ± 5.46 | 30.0 ± 5.31 | 29.75 ± 5.77 | 0.9029 |
| **Race** (N, %) | | | | |
| Black | 10 (33.33) | 4 (28.57) | 6 (37.5) | |
| White | 18 (60.0) | 10 (71.43) | 8 (50.0) | 0.6237 |
| Asian | 1 (3.33) | 0 (0.0) | 1 (6.25) | |
| Other | 1 (3.33) | 0 (0.0) | 1 (6.25) | |
| **Parity** | | | | |
| Multiparous | 17 (56.67) | 7 (50.0) | 10 (62.5) | 0.3625 |
| Nulliparous | 12 (4.80) * 1 missing | 7 (50.0) | 5 (31.25) | |
| **Apgar Scores**, (Mean, SD) | | | | |
| 1 min | 7.33 ± 1.40 | 7.43 ±1.16 | 7.25 ±1.61 | 0.7336 |
| 5 min | 8.77 ± 0.68 | 8.79 ± 0.58 | 8.75 ± 0.78 | 0.8886 |
| **Mode of birth** | | | | |
| Vaginal | 22 (73.33) | 9 (64.29) | 13 (81.25) | 0.4171 |
| Cesarean | 8 (26.67) | 5 (35.71) | 3 (18.75) | |
| **Newborn birth weight** (Mean, SD), grams | 3361.10 ± 271.70 | 3386.30 ± 293.90 | 3339.10 ± 258.40 | 0.6431 |

* Missing swab due to emergency cesarean.

**Table 2.** Intrapartum results.

| GBS Timing | Total (*n* = 30) | Probiotics (*n* = 14) | Placebo (*n* = 16) | *p* Value |
|---|---|---|---|---|
| 36-week SOC (*n*, %) | | | | |
| Positive | 8 (26.67) | 3 (21.43) | 5 (31.25) | 0.6887 |
| Negative | 22 (73.33) | 11 (78.57) | 11 (68.75) | |
| Intrapartum (*n*, %) | | | | |
| Positive | * 6 (20.0) | 3 (21.43) | * 3 (18.75) | 0.639 |
| Negative | 23 (76.67) | 11 (78.57) | 12 (75) | |
| Rate of change (*n*, %) | | | | |
| No group change | 26 (86.67) | 12 (85.7) | 14 (87.5) | 0.8859 |
| Group change | 4 (13.33) | 2 (12.5) | 2 (14.29) | |
| • Pos to Neg | 3 | 1 | 2 | |
| • Neg to Pos | 1 | 1 | 0 | |

* One intrapartum swab was missed in the placebo group due to precipitous birth.

Figure 2 compares the 36-week and intrapartum GBS results for the subgroup participants in the probiotic and placebo groups. One probiotic group participant met the study definition of residual GBS because they had a negative 36-week GBS culture result and a positive intrapartum GBS result. Therefore, they were not treated according to the ACOG guidelines. All neonates were followed up by pediatricians using routine standards.

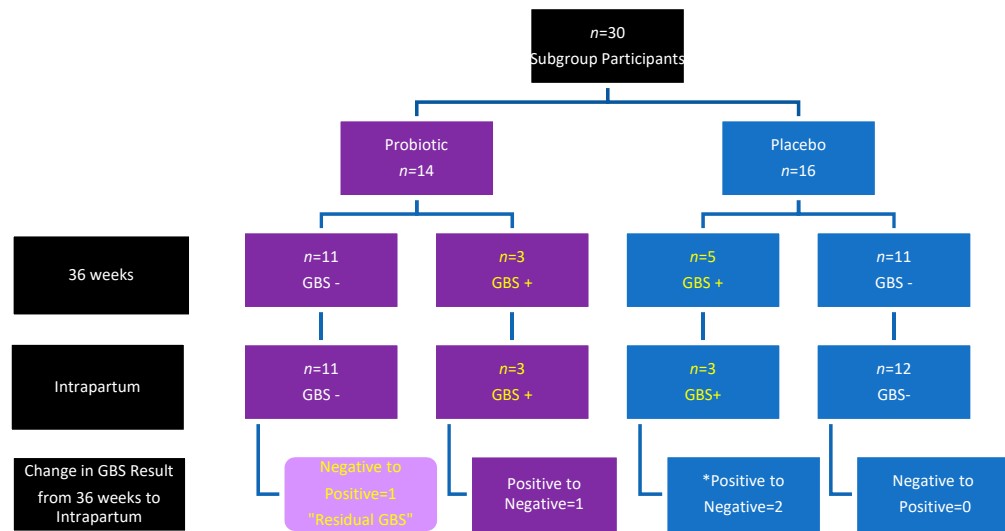

**Figure 2.** Comparison of 36-week and intrapartum GBS results between probiotics and placebo group participants. Note: * one swab missing.

## 4. Discussion

Although it is small, this pilot sub-study is consistent with others showing that 36-to-37-week standard-of-care swabs significantly predicted intrapartum GBS colonization, although mismatched cases did occur. The antepartum and intrapartum GBS findings were mismatched in four cases (4/30, 13.3%), verifying the transitory nature of GBS colonization. Although 27% of the sub-study participants were GBS-positive at 36 weeks, only 20% were GBS-positive at the time of labor and giving birth, perhaps due to the substantial changes the vaginal microbiome undergoes during the third trimester of pregnancy [33].

The antepartum and intrapartum GBS colonization results have been compared in several large descriptive studies. El Helali et al. (2009) [18] conducted a study of 863 pregnant participants and found a rate of discordance between antepartum GBS cultures and intrapartum GBS cultures of 7%. Hussain and colleagues [34] conducted a study of 299 pregnant participants regarding GBS colonization throughout pregnancy with a first trimester urine culture, a 35-to-37-week rectovaginal culture, and an intrapartum rectovaginal culture; they found that the accuracy of antenatal GBS cultures compared to intrapartum cultures was 85% higher, with a sensitivity of 69% and a specificity of 87% [34]. Additionally, 3.7% of the participants who tested negative at 35–37 weeks were GBS-positive intrapartum, and 11% who were GBS-positive at 35–37 weeks were GBS-negative intrapartum [34]. These rates are comparable to the rate in this sub-study (86.67% accuracy; 3.6% tested negative at 36 weeks and positive during labor; 10.7% were GS positive at 36 weeks were GBS-negative intrapartum). However, these larger studies were descriptive and were not a part of a probiotic intervention study. Parente et al. [13] conducted a study of neonates (*n* = 492) with EOGBSD-positive-born parents who screened negative for GBS in the 36-week SOC swabs and estimated that 9% of those who are GBS-negative antepartum converted to GBS-positive at the time when they were in labor and gave birth.

Despite using the same laboratory and analysis methods, in four cases, the participant's antepartum and intrapartum GBS findings were mismatched. Two participants in the placebo group and one in the probiotics group converted from GBS-positive to -negative. Only one case of residual GBS was identified, and this was a probiotic group participant. The potential reasons for mismatched GBS findings include sampling errors (not placing the swab 2 cm into the vagina and 1 cm into the rectum), low GBS colony counts, diet, and alterations in the patient's immune status [13,35,36]. There was no evidence that the probiotic intervention reduced the residual GBS in this small pilot sub-study. More prospective research is needed to determine the clinical significance of this finding. A clinical concern

for residual GBS cases includes a potential increased risk of EOGBSD since those who test negative at 36 weeks are not candidates for intrapartum antibiotic prophylaxis.

No cases of EOGBSD occurred in the parent or pilot sub-study participants, which is consistent with rates reported in the literature. Despite the >30% rate of standard-of-care GBS colonization in the control group participants in this pilot sub-study and the one infant born to a study participant who tested positive during the intrapartum period, no neonatal nasopharynx swabs were positive. More research on the route of neonatal GBS colonization is needed. It is possible that intrapartum antibiotic prophylaxis eliminates GBS bacteria so that it is not detectable in neonates; although, this would not impact the one case of residual GBS who converted to being GBS-positive after 36 weeks. Within just a few minutes of being given the first IAP dose, the fetus is exposed to a high dose of antibiotics [37]. It is also possible that oral/nasopharyngeal swabs are not the correct sites to screen for colonization on neonates. More studies are needed to understand the pathway of neonatal colonization and its relationship with EOGBSD and LOGBSD.

### 4.1. Limitations

The findings of this pilot sub-study are limited by a small sample size of participants in the final year of the parent study. The study topic is limited by the large sample size necessary to identify cases of EOGBSD. We followed up with neonates for 2 months for adverse events, and therefore, this is not long enough to fully evaluate for LOGBSD. In the United States, the current rate of EOGBSD among newborns is 0.22–0.23/1000 [1,4]. At the study site, there were 2517 births in 2017, 2541 in 2018, and 2424 in 2019 (J. Garland, personal communication, 4 February 2020). During the same timeframe (2017–2019), there were two cases of EOGBSD treated in the neonatal intensive care unit (J. Garland, personal communication 4 February 2020). This gives an overall rate of EOGBSD for the facility of 0.27/1000 live births for the years 2017–2019, which is consistent with the US rate of EOGBSD of 0.22/1000 live births [4]. Therefore, a large sample size is needed to identify any cases of EOGBSD at the study site.

A limitation of the parent study is the reported adherence rate to the study probiotic and placebo capsules of only 51–60% percent [21]. This may be because of the recommendation to refrigerate the probiotic and placebo interventions so that the products would not likely be kept with prenatal pills that pregnant people are accustomed to taking daily. Refrigeration may be a barrier to probiotic use in some resource-poor settings, and future trials may need to keep this in mind. Despite the low rate of adherence, the probiotic group participants had significantly more (68%) of the probiotic bacteria contained in the probiotic intervention in their vaginal/rectal microbiomes compared to those of the placebo group participants (32%; $p = 0.04$).

### 4.2. Clinical Implications

EOGBSD prevention presents a challenge to the care of birthing people and their neonates. Midwives and other pregnancy care providers and many of their clients favor judicious use of interventions, such as intravenous lines and antibiotics. For midwives and providers practicing in community settings such as home or birth centers and those practicing in low-resource settings, intrapartum antibiotic prophylaxis is especially challenging. While the universal screening approach with intrapartum antibiotic prophylaxis for those who test positive is currently the best strategy available, it has not eliminated EOGBSD and does not prevent LOGBSD. Further, the universal screening strategy is not agreed upon by all obstetric and midwifery organizations, with organizations such as the Royal College of Obstetricians and Gynecologists (RCOG) recommending a risk-based approach [38]. Efforts concerning antibiotic stewardship in the United States are aimed at reducing routine antibiotic use, and concerns remain about the short- and long-term sequelae of perinatal antibiotic use [39]. In a systematic review and meta-analysis of universal screening versus risk-based screening, researchers found fewer EOGBSD cases in the universal screening group and did not clearly identify the increased antibiotic use in the screening group [40].

GBS has been identified by the World Health Organization (WHO) as a priority, particularly because EOGBSD disproportionately impacts infants in underdeveloped and developing nations [41]. While there are promising developments for a vaccine against GBS, the vaccine is not close to market readiness [42,43]. Further studies of the vaccine will require a large sample of pregnant people. Following the rapid development of mRNA COVID-19 vaccines, vaccine development may be a promising primary prevention method. These issues highlight the importance of continued study of other means of primary prevention of both EOGBSD and LOGBSD. The use of probiotic interventions aimed at a reduction of GBS colonization may be one part of a larger primary prevention strategy to reduce both intrapartum antibiotic prophylaxis use, GBS carriage, and EOGBSD.

### *4.3. Feasibility*

Importantly, this midwifery-led, pilot, exploratory study shows the feasibility and acceptability of collecting intrapartum samples. The midwives who collected the intrapartum swabs are part of large midwifery group that provides obstetrics triage services within the hospital facility. The midwives have other clinical responsibilities, including the management of intrapartum patients. Despite this, only one maternal intrapartum swab out of thirty was missed during a precipitous birth. The midwives were able to conduct clinical research and integrate swab collection into their clinical responsibilities. Two neonatal swabs were missed, one after the precipitous birth, and one following an unplanned cesarean section. No participants declined intrapartum swabs.

While the pilot sub-study demonstrated the feasibility of intrapartum and neonatal GBS data collection, it may not be a data collection model that could be sustained through a multi-year, multi-site, double-blind, randomized, placebo-controlled trial of probiotics to reduce GBS. Although it is feasible, the pilot sub-study highlighted that antepartum GBS significantly predicted the patient's intrapartum GBS status. Certain aspects of intrapartum specimen collection remain unknown, including the influence of amniotic fluid and/or blood on GBS culture collection. If researchers wish to include an intrapartum GBS culture as an outcome, it is recommended that these issues are addressed. The use of a polymerase chain reaction (PCR) test may be more accurate and could be used in combination with culturing. Intrapartum swabs are likely only useful if a quantitative PCR methodology is available and/or to guide neonatal care.

Future sufficiently powered, well-controlled studies of antenatal probiotic interventions with enhanced adherence strategies are needed. An analysis of the effect of probiotic interventions on the various GBS serotypes would provide information about probiotic efficacy and why some participants respond and others do not.

### 5. Conclusions

The findings of this study verified that 36-week standard-of-care GBS screening significantly predicts patients' intrapartum GBS statuses. Although this study has a small sample size, there was no evidence that the probiotic intervention reduced the residual GBS. Midwives, physicians, and pregnancy care providers should continue to follow the ACOG recommendations for EOGBSD prevention, while scientific efforts continue to search for effective primary prevention strategies. While intrapartum and neonatal GBS swabs were acceptable and feasible for both midwives and intrapartum participants, it is unclear if this measure is useful. More studies are needed about the efficacy of probiotics against GBS and to understand the route of neonatal GBS colonization and the progression to EOGBSD. Based on the findings of the parent study, midwives can consider offering specific probiotic interventions to their clients to reduce the GI symptoms of pregnancy.

**Author Contributions:** Conceptualization, L.H. and L.V.; methodology, L.H. and L.V.; formal analysis, M.S., L.H., L.V. and E.M.; investigation, E.M., L.H., L.V. and M.F.; data curation, M.S.; writing—original draft preparation, E.M. and L.H.; writing—review and editing, M.F., L.V., M.S. and K.R.; supervision, L.H.; project administration, L.H., E.M. and M.F.; funding acquisition, L.H. and L.V. All authors have read and agreed to the published version of the manuscript.

**Funding:** This research was funded by a Victoria Wallace Research Award, Marquette University College of Nursing. Research reported in this publication as "parent study" was supported by Eunice Kennedy Shriver National Institute of Child Health and Human Development of the National Institutes of Health under award number R21HD095320.

**Institutional Review Board Statement:** The study was conducted in accordance with the Declaration of Helsinki, and approved by the Institutional Review Board of Aurora Health Care protocol #17-136 on 20 February 2019.

**Informed Consent Statement:** Not applicable.

**Data Availability Statement:** The data presented in this study are available on request from the corresponding authors. The data are not publicly available because this was an exploratory sub-study.

**Acknowledgments:** The authors would like to thank Diana Kleber, RN, the research coordinator, the nurses and midwives at Aurora Sinai Medical Center, and the pregnant people and newborns who participated in the study.

**Conflicts of Interest:** American Lifelines, the makers of Florajen3, provided probiotic and placebo products for this study. This company had no role in the design of the study; in the collection, analyses, or interpretation of data; in the writing of the manuscript; or in the decision to publish the results. The authors declare no conflict of interest.

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
