# Peer review of "Exploring the Feasibility of Intrapartum GBS Collection to Identify Residual GBS in a Pilot Study of an Antenatal Probiotic Intervention"

_2673-8007, doi:10.3390/applmicrobiol3030052_

Round 1

Reviewer 1 Report

Review of manuscript

Exploring the Feasibility of Intrapartum GBS Collection to  Identify Residual GBS in a Pilot Study of an Antenatal Probiotic Intervention

Dear authors, the manuscript is well written and the topic is somewhat relevant, despite no conclusion can be achieved with probiotic use. I had done some suggestions and made some questions that can be found below.

L 10 and 11: “collection” could be changed for “colonization”.

L 18 and 22: they have exactly the same sentence “Antepartum GBS results significantly predicted intrapartum results”. In addition, we do not know what this information means. I suggest authors be clearer in such sentence, for instance, showing colonization rates in both antepartum and intrapartum screening.

L 24-25: I suggest changing the order of this sentence “Neonatal oral and nasopharyngeal swabs did not recover GBS” to “GBS was not recovered from…

L 31: As far as I know, GBS is not a leading cause of maternal mortality. I suggest authors correct this information.

L 50-53: Besides colonization at birth, there is also the intrauterine infection, that can be associated with EOD. I suggest the authors mention it.

L 55: “person” could be changed for “woman”.

L 58-59: I suggest reorganizing the sentence, eg. “Without intrapartum antibiotic prophylaxis, 50% of the neonates born from GBS positive mothers will be colonized by GBS and 1-2% will develop…”

L 68-70: “LOGBSD presents between 7 days of life and 2-3 months “. Actually, LOD has been considered until 3 months, not 2-3 months. Line 70 has the same information, so I suggest removing the first sentence, L 68.

L 82: “people” could be changed for “women”.

L 100: Authors should be more explicit about probiotic mechanisms of action, eg. acidification of what? Is bacterial adhesion impaired or enhanced?   

L 107-108: correct - “systematic review and meta-analysis of ten studies, (five in vitro studies and six clinical trials)”

L 113: RCTs have not been mentioned before, so, write the meaning here.

L 139-140: correct - “the sample size of 83 (39 probiotic and 43 placebo).”

Tables 1 and 2, Figure 2: I suggest authors present data of the subjects that effectively were enrolled in this sub-project – one participant had a precipitous birth and swab was missing, so, her data from 36 weeks should not be part of the analysis/results.

L 218-219: correct “they” – is only one participant.

L 229: “GBS findings were mismatched in 4 cases…” authors should state that one participant was missed in intrapartum screening, so, mismatched occurred in 3 cases.

L 244: correct - “GS”.

L 247: “born parents who screened…” preposition “from” is missing.

L 250-251: again, authors should consider reviewing “four cases” of mismatch; “Two participants in 251 the placebo group…”

L 261: Did positive GBS parturients at week 36 receive IAP, independent of being placebo or probiotic group? If so, I suggest include such information since it explains, in part, no cases of EOD.

L 289: correct English grammar “may to need”

L 334-335: I suggest authors stress the usefulness of intrapartum swabs only if quantitative PCR methodology is available at the clinical laboratory.

L 347: “intrapartum and neonatal GBS swabs were acceptable and feasible for both midwives and intrapartum participants, it is unclear if this measure is useful”. Maternal intrapartum swabbing is useful within the above conditions, ie, the availability of molecular techniques of diagnosis.

Few English errors were found. They are stressed about my comments.

Author Response

Reviewer #1

Dear authors, the manuscript is well written, and the topic is somewhat relevant, despite no conclusion can be achieved with probiotic use. I had done some suggestions and made some questions that can be found below. Thank you. We are very grateful for your time and suggestions to improve the manuscript.

L 10 and 11: “collection” could be changed for “colonization”. Done, also changed sentence slightly to improve clarity.

L 18 and 22: they have exactly the same sentence “Antepartum GBS results significantly predicted intrapartum results”. In addition, we do not know what this information means. I suggest authors be clearer in such sentence, for instance, showing colonization rates in both antepartum and intrapartum screening. Changed sentence in Line 18 to add percentage. Removed sentence on Line 22 and combined with another sentence in the conclusion to improve clarity.

L 24-25: I suggest changing the order of this sentence “Neonatal oral and nasopharyngeal swabs did not recover GBS” to “GBS was not recovered from… Done, thank you.

L 31: As far as I know, GBS is not a leading cause of maternal mortality. I suggest authors correct this information. Removed “maternal”. Our understanding is that GBS is a cause of significant maternal morbidity rather than mortality. This is from: Brokaw A, Furuta A, Dacanay M, Rajagopal L, Adams Waldorf KM. Bacterial and Host Determinants of Group B Streptococcal Vaginal Colonization and Ascending Infection in Pregnancy. Front Cell Infect Microbiol. 2021;11:720789. Published 2021 Sep 3. doi:10.3389/fcimb.2021.720789 and their citation of:

Hall, J., Adams, N. H., Bartlett, L., Seale, A. C., Lamagni, T., Bianchi-Jassir, F., et al. (2017). Maternal Disease With Group B Streptococcus and Serotype Distribution Worldwide: Systematic Review and Meta-Analyses. Clin. Infect. Dis. 65, S112–S124. doi: 10.1093/cid/cix660

L 50-53: Besides colonization at birth, there is also the intrauterine infection, that can be associated with EOD. I suggest the authors mention it. Intraamniotic infection and endometritis are mentioned in Lines 36-37. Respectfully, does the reviewer suggest we also mention it here?

L 55: “person” could be changed for “woman”. Thank you for this suggestion. We used the recommendation of American College of Obstetrician/Gynecologists and American College of Nurse-Midwives for gender inclusive language regarding care of pregnant individuals: https://www.acog.org/clinical-information/policy-and-position-statements/statements-of-policy/2022/inclusive-language

Editor: please clarify the preference of this journal and we will make appropriate changes throughout the document.

L 58-59: I suggest reorganizing the sentence, eg. “Without intrapartum antibiotic prophylaxis, 50% of the neonates born from GBS positive mothers will be colonized by GBS and 1-2% will develop…” Done, thank you.

L 68-70: “LOGBSD presents between 7 days of life and 2-3 months “. Actually, LOD has been considered until 3 months, not 2-3 months. Line 70 has the same information, so I suggest removing the first sentence, L 68. Done, thank you.

L 82: “people” could be changed for “women”. As above.

L 100: Authors should be more explicit about probiotic mechanisms of action, eg. acidification of what? Is bacterial adhesion impaired or enhanced? Done, thank you.

L 107-108: correct - “systematic review and meta-analysis of ten studies, (five in vitro studies and six clinical trials)” This is correct, one of the clinical trials included both an ‘in vitro’ and an ‘in vivo’ arm, so it is counted twice in the trial. Corrected to read… five in vitro studies and six clinical trials including one of the in vitro studies. Here is the citation: Hanson L, VandeVusse L, Malloy E, et al. Probiotic interventions to reduce antepartum Group B streptococcus colonization: A systematic review and meta-analysis. Midwifery. 2022;105:103208. doi:10.1016/j.midw.2021.103208

L 113: RCTs have not been mentioned before, so, write the meaning here. Done, thank you.

L 139-140: correct - “the sample size of 83 (39 probiotic and 43 placebo).” Corrected, thank you.

Tables 1 and 2, Figure 2: I suggest authors present data of the subjects that effectively were enrolled in this sub-project – one participant had a precipitous birth and swab was missing, so, her data from 36 weeks should not be part of the analysis/results. Thank you for this feedback. We used an intention to treat model, so our statistician recommended that the missed cases are counted in our total. Regression was done on the 29 maternal swabs, and the other analyses were done on the non-missing swabs.

L 218-219: correct “they” – is only one participant. Thank you, removed the word “they.” The use of the word “they” in the US is intended to take the place of gendered pronouns he/she. We understand that this does not read clearly.

L 229: “GBS findings were mismatched in 4 cases…” authors should state that one participant was missed in intrapartum screening, so, mismatched occurred in 3 cases. 3 participants covered from positive to negative (1 in the probiotics group and 2 in the placebo group), and 1 participant converted from negative to positive (in the probiotics group) for a total of 4 mismatched cases. The missing swab is not included in the cases of mismatch.

L 244: correct - “GS”. Done, thank you.

L 247: “born parents who screened…” preposition “from” is missing. Corrected, thank you.

L 250-251: again, authors should consider reviewing “four cases” of mismatch; “Two participants in 251 the placebo group…” We apologize, but unsure what the reviewer means here. There are four cases of mismatch. These do not include the missed swabs.

L 261: Did positive GBS parturients at week 36 receive IAP, independent of being placebo or probiotic group? If so, I suggest include such information since it explains, in part, no cases of EOD. This is from Line 270-272, which I think answers the reviewer’s recommendation: It is possible that the intrapartum antibiotic prophylaxis eliminated GBS bacteria so that it was not detectible in neonates, although this would not impact the one case of residual GBS who converted to GBS positive after 36 weeks.

L 289: correct English grammar “may to need” Corrected, thank you.

L 334-335: I suggest authors stress the usefulness of intrapartum swabs only if quantitative PCR methodology is available at the clinical laboratory. Added a sentence for clarity about lack of utility intrapartum.

L 347: “intrapartum and neonatal GBS swabs were acceptable and feasible for both midwives and intrapartum participants, it is unclear if this measure is useful”. Maternal intrapartum swabbing is useful within the above conditions, ie, the availability of molecular techniques of diagnosis. Clarified, thank you.

Reviewer 2 Report

In this manuscript authors aimed to explore the feasibility of intrapartum maternal Group B Streptococcus (GBS) collection and immediate post-birth neonatal GBS collection for use in a larger trial; and to identify cases of residual GBS, hypothesized to be less in the probiotics group. THEIR RESULTS showed that 36 to 37 week standard of care swabs significantly predicted intrapartum GBS colonization, although mismatched cases do occur.

Please state the aim of the study in introduction section.

The introduction is too long and should be shortened.

Lines 148-154 referred to aim of the study and should be removed and included in the introduction section.

In the result section, the information regarding the management and follow up of neonates for screening / prevention of EOS was messed.

Line 209-210 the results should not be limited to p value only but information about the results should also be given (coefficient β, CI 95%)

Although studies report that the rates of EOGBS disease were approximately 4-fold higher than those of LOGBS the fact that the study was limited only to EOGBS and no information was given regarding LOGBS should be mention as another limitation of the study.

Lines 299-300 I am not sure for this in a recent meta analysis conducted by Hasperhoven et al (Hasperhoven GF, Al-Nasiry S, Bekker V, Villamor E, Kramer B. Universal screening versus risk-based protocols for antibiotic prophylaxis during childbirth to prevent early-onset group B streptococcal disease: a systematic review and meta-analysis. BJOG. 2020;127(6):680-691. doi:10.1111/1471-0528.16085) authors concludet that: “screening protocols were associated with lower rates of EOGBS disease compared with risk‐based protocols. While there is insufficient evidence to assume that risk‐based policies reduce the use of prophylactic intrapartum antibiotics, these protocols might not be able to protect infants from EOGBS disease to the same extent as general screening does. These findings can be of help to future policy‐making and individual pregnancy counselling”

Author Response

Reviewer #2

Please state the aim of the study in introduction section. Done, thank you. Moved a paragraph up from materials and methods.

The introduction is too long and should be shortened. Shortened intro and added a “Background” section.

Lines 148-154 referred to aim of the study and should be removed and included in the introduction section. Done, thank you.

In the result section, the information regarding the management and follow up of neonates for screening / prevention of EOS was messed. Added a sentence explaining this.

Line 209-210 the results should not be limited to p value only but information about the results should also be given (coefficient β, CI 95%) Added (OR=33.33, 95% CI=[2.8, 392.5], p=0.005), thank you.

Although studies report that the rates of EOGBS disease were approximately 4-fold higher than those of LOGBS the fact that the study was limited only to EOGBS and no information was given regarding LOGBS should be mention as another limitation of the study. Added this limitation, thank you.

Lines 299-300 I am not sure for this in a recent meta analysis conducted by Hasperhoven et al (Hasperhoven GF, Al-Nasiry S, Bekker V, Villamor E, Kramer B. Universal screening versus risk-based protocols for antibiotic prophylaxis during childbirth to prevent early-onset group B streptococcal disease: a systematic review and meta-analysis. BJOG. 2020;127(6):680-691. doi:10.1111/1471-0528.16085) authors concludet that: “screening protocols were associated with lower rates of EOGBS disease compared with risk‐based protocols. While there is insufficient evidence to assume that risk‐based policies reduce the use of prophylactic intrapartum antibiotics, these protocols might not be able to protect infants from EOGBS disease to the same extent as general screening does. These findings can be of help to future policy‐making and individual pregnancy counseling” Thank you for sharing reference, added it and sentence with the author’s conclusion.

Round 2

Reviewer 2 Report

The authors  have responded to comments accurately and the manuscript was improved.